# A synthetic tubular molecular transport system

Pierre Stömmer[1], Henrik Kiefer[2], Enzo Kopperger [3], Maximilian N. Honemann [1], Massimo Kube[1], Friedrich C. Simmel[3], Roland R. Netz [2] & Hendrik Dietz [1✉]

Creating artificial macromolecular transport systems that can support the movement of molecules along defined routes is a key goal of nanotechnology. Here, we report the bottom-up construction of a macromolecular transport system in which molecular pistons diffusively move through micrometer-long, hollow filaments. The pistons can cover micrometer distances in fractions of seconds. We build the system using multi-layer DNA origami and analyze the structures of the components using transmission electron microscopy. We study the motion of the pistons along the tubes using single-molecule fluorescence microscopy and perform Langevin simulations to reveal details of the free energy surface that directs the motions of the pistons. The tubular transport system achieves diffusivities and displacement ranges known from natural molecular motors and realizes mobility improvements over five orders of magnitude compared to previous artificial random walker designs. Electric fields can also be employed to actively pull the pistons along the filaments, thereby realizing a nanoscale electric rail system. Our system presents a platform for artificial motors that move autonomously driven by chemical fuels and for performing nanotribology studies, and it could form a basis for future molecular transportation networks.

[1] Lehrstuhl für Biomolekulare Nanotechnologie, Physik Department, Technische Universität München, Munich, Germany. [2] Fachbereich Physik, Freie Universität Berlin, Berlin, Germany. [3] Lehrstuhl für Physik Synthetischer Biosysteme, Physik Department, Technische Universität München, Munich, Germany. ✉email: dietz@tum.de

Transporting matter along one-dimensional tracks instead of arbitrary trajectories offers efficiency advantages. This notion holds true on the macroscale and also for molecular-scale transport in liquid solution. Eukaryotic cells have evolved a sophisticated cellular transportation system in which motor proteins move with micrometer long travel ranges and μm/s displacement speeds along a variety of cellular filaments[1–4]. Creating similarly efficient artificial means of transporting molecules is an unmet challenge for nanoscale science and technology. There are several fundamental aspects that can guide the design of suitable tracks and means to attach particles to them so that the translational degree of freedom along the tracks is retained. For example, the structure of molecular tracks defines the free energy landscape that directs the diffusive motion of attached particles. The barriers in this landscape control the mobility in a Boltzmann-weighted fashion, where energetic barriers larger than a few units of thermal energy ($k_BT$) can no longer be easily overcome by thermal fluctuations and thus represent roadblocks. Furthermore, due to Brownian motion in liquid solution, a particle moving on a molecular track must be tightly attached to it at all times or it will diffuse away. Natural sliding-clamp proteins[5] are mechanically interlocked with their tracks in a ring-on-axle fashion (Fig. 1a left), while natural molecular motors such as kinesin or myosin typically walk "on" their filamentous track, thereby realizing a form of multivalent attachment with alternating bond formation (Fig. 1a right). In principle, mechanical interlocking of a ring-like object on an axle should provide the highest mobility, because displacements of the ring do not necessarily require the breaking and reforming of molecular bonds. In the case of a molecular walker, by contrast, molecular bonds need to break and re-form repeatedly. Therefore, there will be a trade-off between particle mobility and the risk of losing the particle to solution. There are also biological examples of

molecular transport inside tubes[6–9]. For instance, bacterial secretion systems involve long protein channels through which molecular transport is driven by ATP-consuming motors (typically AAA + ATPases). In the case of Type IV secretion systems (T4SS) involved in bacterial conjugation[6], DNA and proteins are transported from one bacterial cell through a long pilus to another cell. Transport inside hollow tubes has several beneficial properties: transport is insulated from the environment which allows rapid movements in the absence of exterior disturbances (as exemplified in subways and highspeed trains also on the macroscale) and the particles moving through the tubes cannot diffuse away.

Here, we used programmable self-assembly with DNA to create an artificial molecular transport system that reproduces several of the attractive properties of natural molecular-scale transporters, including multiple-micrometer-long travel ranges and μm/s displacement speeds. DNA nanotechnology has been previously employed to construct a variety of artificial molecular devices and machines[10–12]. 3D DNA components have been designed and put together to create pivots, hinges, crank sliders, and rotors[13–15], in which DNA strand linkages or particular design features, such as mechanically interlocked but not directly connected parts constrain the range of movements of these devices. The utilization of strand displacement reactions (SDR)[10,13,15,16] has been pivotal in allowing DNA nanoengineers to dynamically reconfigure DNA nanostructures. SDR-based DNA walkers have been created that can move on various types of linear tracks or 2D surfaces[17–20]. Natural enzymes such as polymerases and nucleases have also been coupled to SDR-based walkers to bias their movements[21,22]. Because displacements of SDR walkers require breaking and reforming double-helical domains formed between walker and track, the speed of SDR walkers is coupled to the kinetics of these reactions, which can limit overall performance. Li et al addressed

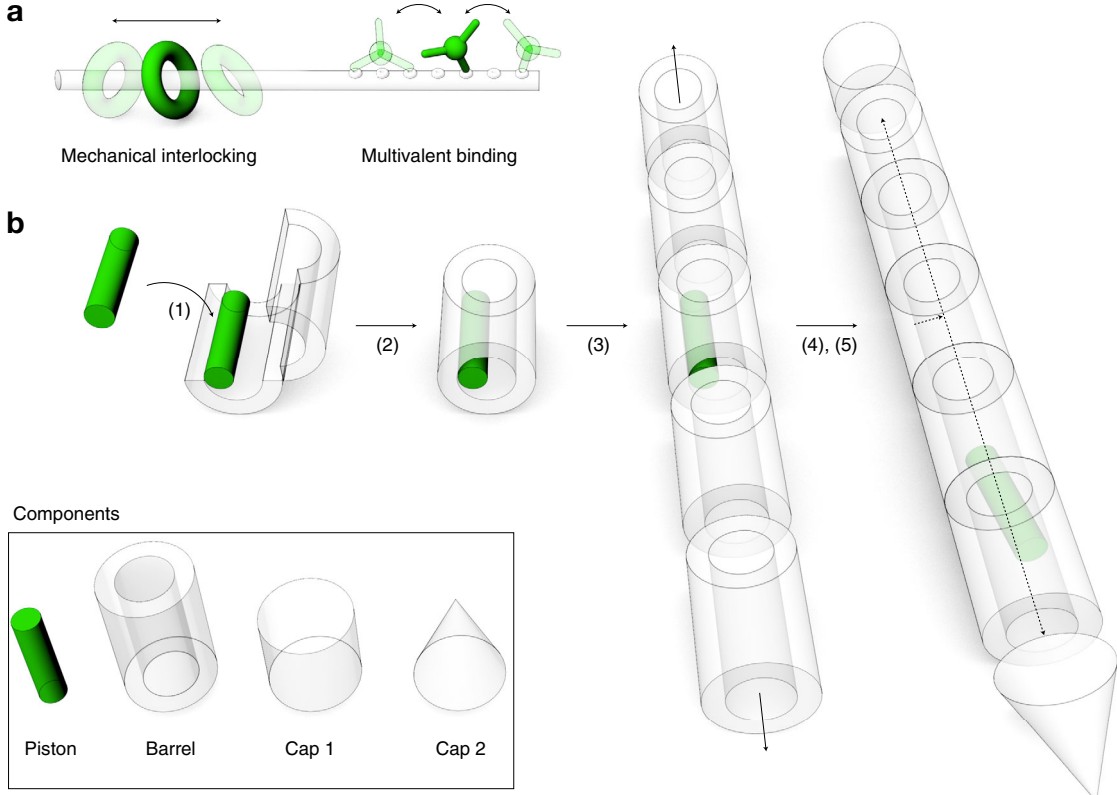

**Fig. 1 Concept for constructing a molecular hyperloop. a** Strategies for confining a mobile random walker onto a one-dimensional (1D) track. **b** Schematics of inverted mechanical interlocking and stepwise self-assembly of such a system.[1] loading the "piston" building block onto the barrel building block in open state,[2] closure of barrel,[3] polymerization of barrel building blocks into long filaments,[4] capping of filaments,[5] release of piston from docking site.

this challenge recently with a sequence-optimized cartwheeling DNA object capable of undergoing undirected diffusive motions on a two-dimensional DNA carpet with the hitherto fastest reported diffusive mobilities of DNA objects up to 17 nm²/s[23]. These diffusivities are still many orders of magnitudes lower than those reported for natural motor proteins, which may be attributed to limitations arising from the kinetics of DNA strand hybridization and dissociation.

## Results

**Design and validation of individual components**. To overcome the motility limitations in previously reported random walker designs, we decided to use inverted mechanical interlocking to realize our molecular transport system: instead of a ring-on-an-axle, our mobile unit takes the form of a "piston" that can move along a hollow tube-like filament (Fig. 1b). To build this system, we load the piston into a subunit of the tube called "barrel", extend the barrel on both sides by polymerization into a long tube using empty barrels, followed by capping of the terminal openings of the tube to prevent the mobile unit later from exiting the track, and finally we release the piston from its docking site in the tube. To facilitate loading the piston into the barrel, we first load the piston into an opened-up conformation of the barrel, followed by closing this object.

We used multi-layer DNA origami[24] and docking schemes[25] to build and attach to each other the components of our system (Fig. 2). The piston, barrel, and the two capping units (Supplementary Figs. 1–7) are each comprised of 10, 82, and 96 helices, respectively, arranged in honeycomb patterns. The piston has a simple rod-like shape and is folded from a 1033 bases-long custom-sequence scaffold strand[26]. It is ~40 nm long and has a cross section of ~8 by 12 nm. The barrel has a hexagonal shape and is folded from two 7560 bases-long orthogonal-sequence scaffold strands[26] in a one-pot folding reaction. We also created a hinge mechanism that divides the barrel into two half tubes, connected by flexible single-stranded parts. The barrel is ~64 nm long and has an inner tube and outer tube diameter of ~15 and ~30 nm, respectively. The piston is therefore only slightly thinner than the central bore of the barrel. Based on this design, we expect that the motion of the piston will be effectively constrained to two degrees of freedom: translation along the barrel axis and rotation around the long axis of the piston. The two caps are each folded from 7560 bases long scaffold strands. They have arrowhead-like shapes and are 32 and 42 nm long, respectively, with a diameter of ~30 nm. We validated the successful assembly of all components by direct imaging with negative-staining TEM (Fig. 2) and by gel-electrophoretic mobility analysis (EMA) (Supplementary Fig. 8).

**Assembling the molecular transport system**. The first construction step entailed loading the piston monomer into the opened-up barrel monomer. The piston-barrel interaction must endure through subsequent steps such as barrel-closure and track polymerization, but it must also be fully reversible to allow for releasing the piston from the docking site inside the extended tunnel. To satisfy these criteria, we tested several piston variants (Supplementary Fig. 9) and different docking strategies (Supplementary Fig. 10) while iterating through the cycle of piston-loading, track assembly, and release in extended tracks. To dock the piston to the barrel, we created a protrusion on the piston variants which is shape-complementary to a recess located in the interior surface of the barrel (Fig. 2c). We used strand hybridization of single-stranded overhangs and scaffold loops to link the edges of the protrusion of the piston to those of the recess in the barrel (Fig. 2c, Supplementary Fig. 11 design). Piston-barrel

assembly was validated by EMA and by negative-staining TEM (Fig. 2c, right). The second construction step is to close up the barrel (Fig. 2d). To this end, the two half tubes feature a second set of shape-complementary protrusions and recesses on the tube edges. The interfaces of these features are comprised of single-stranded scaffold loops. Upon addition of sequence-complementary oligonucleotides, the barrel is then permanently closed by strand hybridization bridging protrusion and recess interfaces, which we validated by EMA and negative-staining TEM (Fig. 2d, Supplementary Figs. 12 and 13).

The third step entailed polymerizing the piston-loaded barrel (Fig. 2d) with empty barrel monomers (Fig. 2e) to build long tracks (Fig. 3a). To this end, we tested a panel of helical interface interaction designs and a variety of reaction protocols. We eliminated variants that gave only short multimers, yielded branched filament networks instead of single filaments, and had the tendency to polymerize not only in the designed head-to-tail but also in head-to-head or tail-to-tail configurations, thereby causing a constriction in the central bore that can block the mobile unit from diffusing along the track (Supplementary Fig. 14). In the final design solution, our track polymerization was performed by stepwise addition of two sets of oligonucleotides: (1) sequence complementary to scaffold loops at the helical interface of only one of six hexagonal facets of the barrel, (2) sequence complementary to the barrel scaffold loops of the remaining five facets of the barrel (Supplementary Fig. 15). In the fourth assembly step, we sealed the terminal openings of the tunnels by adding the capping building blocks. The caps are attached via single-stranded overhangs at the cap ends that hybridize to the helical interfaces at the ends of filaments. EMA and negative-staining TEM confirmed that the capping reactions worked as desired (Fig. 3a, Supplementary Figs. 16, S17, and S18).

As a result, we obtained capped, piston-containing, multiple-micrometer-long filaments that appeared mostly straight with few kinks and without other obvious defects as seen by negative-staining TEM (Fig. 3b, c, Supplementary Fig. 19). The caps can be discerned in the TEM images (Fig. 3b, insets). We labeled the barrel and piston monomers with cyanine-5 and cyanine-3 fluorophores (Supplementary Figs. 20 and 21), respectively, to allow imaging by fluorescence microscopy. We also labeled the barrels with biotin moieties along a six-helix-bundle-shaped bulge running along the tunnel to immobilize the filaments on neutravidin-coated surfaces (Supplementary Fig. 22). The fluorescence-microscopy images that we acquired from these samples (Fig. 3c, Supplementary Movie 1) are reminiscent of images known from motility assays with natural motor proteins and their filaments[4].

The last step in the construction of our molecular transport system is releasing the piston from its docking site in the central bore of a fully assembled and capped track. During the early iterations of designing our system (Supplementary Fig. 10), the piston monomer had very low mobility for multiple reasons, so it was difficult to judge whether the release was successful by single-particle tracking in real time. We therefore resorted to monitoring efflux of pistons from barrels and tracks with EMA, negative-staining TEM, and fluorescence microscopy. That is, we acquired images from piston-loaded but uncapped tracks prior and after subjecting the samples for several hours or days to putative piston-releasing conditions (Supplementary Fig. 23). Using this strategy, we identified procedures that successfully caused piston release in situ, even though the actual diffusive mobility was too low to be seen in real-time at that point. In our final design solution, we released the piston from its docking site inside the capped tracks by adding invader strands from the outside. The invader strands permeate through the interhelical cavities in the filament walls and release the piston by toehold mediated strand

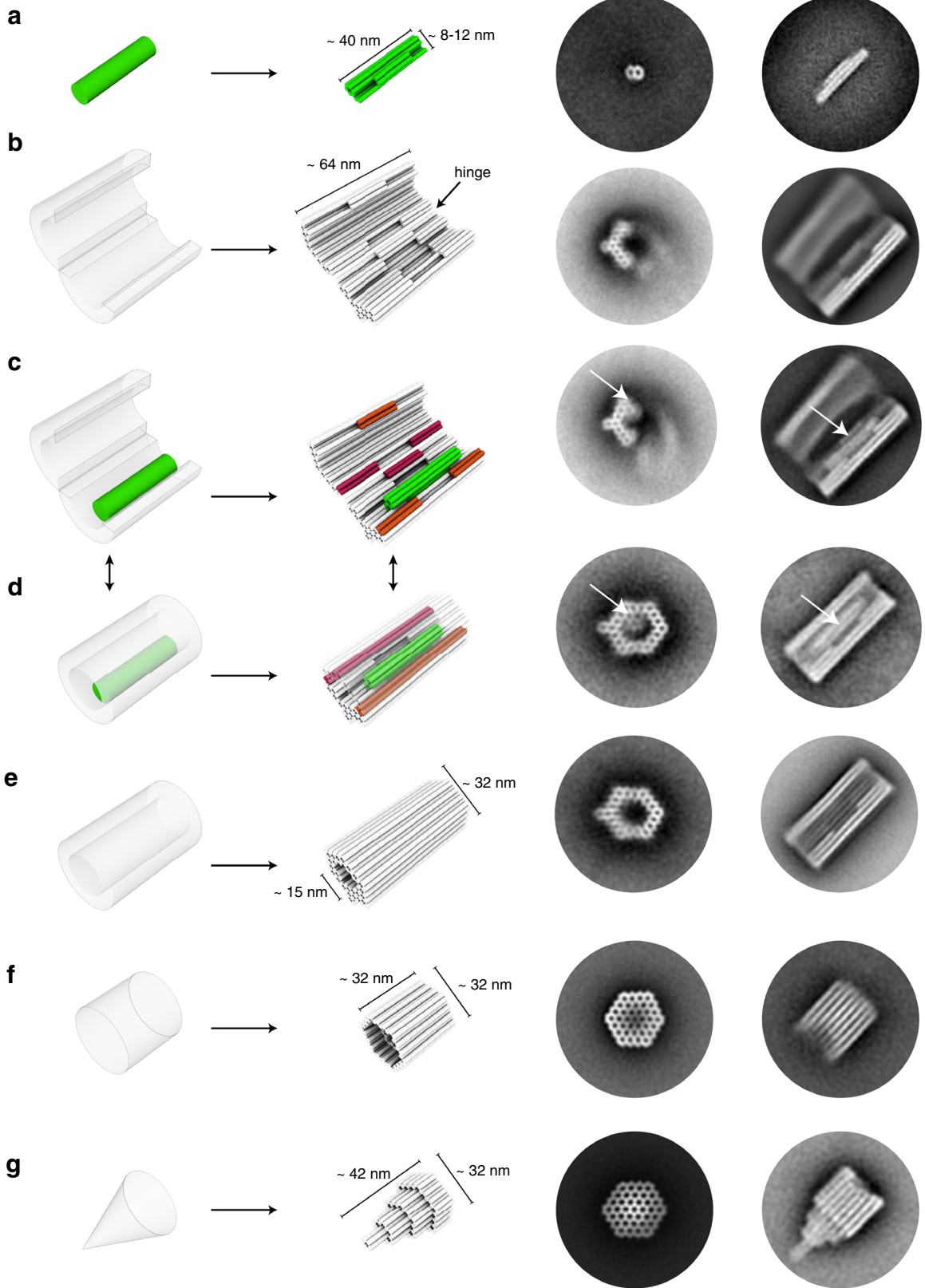

**Fig. 2 Construction of components with DNA origami. a–g** Left: Schematics of how target components are approximated as cylinder models. Cylinders represent DNA double-helices. Right: Representative 2D class average images from negative-staining TEM micrographs, with viewing angle along the helical axis and perpendicular to the helical axis, respectively. Scale bar for all class averages: 20 nm. **c, d** Orange, magenta: protrusions and recesses involved in closing the barrel. Source data are provided as a Source Data file.

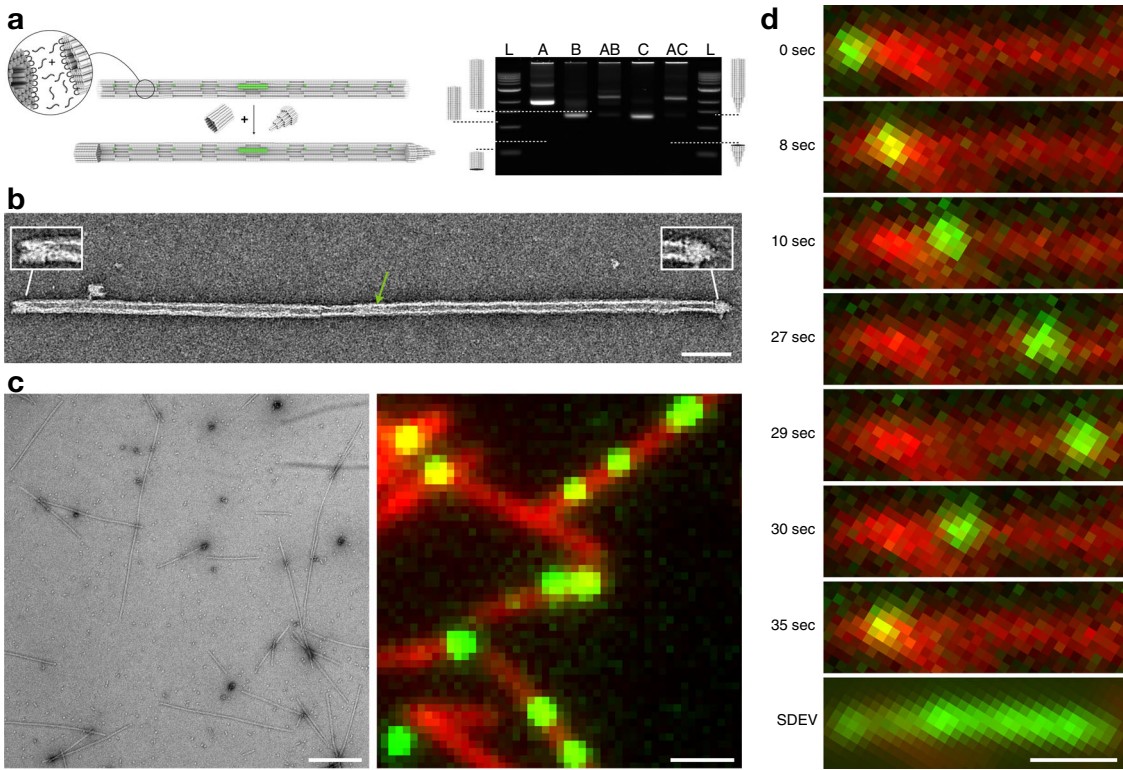

**Fig. 3 Filament polymerization, capping, and single-particle imaging. a** Left: Schematic illustration of the polymerization reaction and subsequent capping of the filament ends. Filament polymerization is induced by addition of DNA oligonucleotides that bridge barrel monomer helical interfaces. Subsequently, capping objects are added to solution which feature single-stranded DNA overhangs complementary to barrel ends. Cap attachment quenches further polymerization. The piston (green object inside the filament) is now sterically trapped. Right: Laser-scanned image of an agarose gel (2%, 21 mM MgCl$_2$, 90 V, 90 min, ice-water bath) on which the following samples were electrophoresed: L = 1 kb ladder, A = closed barrel, B = capping object, AB = closed barrel-capping object 1 dimers, C = closed barrel-capping object 1 dimers, C = capping object 2, AC = closed barrel-capping object 2 dimers. The total number of conducted gel-electrophoretic analyses of samples prepared following the same protocol as illustrated here (steps 10 and 11 in the transport system sample preparation protocol) was 3 times and showed similar results. **b** Exemplary negative-staining TEM image of a capped filament. Green arrow: piston. Insets highlight cap monomers. Scale bar: 100 nm. The total number of conducted TEM analyses of samples prepared following the same protocol as illustrated in the transport system sample preparation protocol was 5 times and showed similar results. **c** Left: Typical field of view negatively stained TEM image of polymerized filaments. Right: Typical field of view TIRF image of polymerized filaments with piston objects trapped inside. The filaments are labeled with Cyanine-5 dyes (10 per barrel monomer), the piston carries 8 Cyanine-3 dyes, image is merged from the two fluorescence channels. Scale bars: 1 μm. The total number of conducted TEM analyses of samples prepared following the same protocol as illustrated in the transport system sample preparation protocol exceeded 10 times and showed similar results. **d** Exemplary sequence of single frames taken from a TIRF movie reflecting movement of a piston along a filament. Bottom: Standard deviation from the mean image for the entire movie (6000 frames, frame rate = 10/s), illustrating that the piston has traveled across the entire length of this ~3 μm long filament. Scale bar: 1 μm. Source data are provided as a Source Data file.

displacement (Supplementary Fig. 24). Most pistons within a field of view were mobile within several minutes of incubation with the invader strands at room temperature.

**Real-time observation of piston movements.** With thus prepared samples, we observed many filaments with piston units performing random diffusive motions along the tracks they were constrained on (Fig. 3d, Supplementary Movie 1, Supplementary Figs. 25–36). The number of mobile units per track can be controlled via the initial stoichiometry of piston-loaded barrels to barrels for polymerizing multimers. Hence, situations can be created where multiple pistons move on the same track, leading to situations where they apparently bump into each other, move together for a while, and then part ways (Supplementary Movies 2 and 3).

We quantitatively analyzed the motions of the mobile units on their tracks using super-resolution centroid tracking[27], which yielded position over time trajectories (Fig. 4a, b, Supplementary Figs. 25–36). The pistons featured multiple fluorophores, which enabled continuous particle tracking typically over time spans

around ~10 min before the signal got too dim because of dye bleaching.

We observed diffusive motions of single pistons along the entire length of the underlying filaments. The farthest motions we recorded occurred over a total length of 3 μm (Fig. 4a, Supplementary Movie 4). There was heterogeneity with respect to the diffusive mobility of the particles. Some particles would get stuck repeatedly at conserved sites (e.g., Fig. 4a) which we attribute to localized roadblocks in the tracks. Such defects could be caused by single-stranded sites or slightly angled connections between barrel monomers, which would require an energetically unfavorable bending deformation of the piston in order to move through such a constriction.

Other particles did show very high mobility, moving over micrometer distances within fractions of seconds without getting stuck (Fig. 4b, Supplementary Movie 5). We used the single-particle position-time traces to compute the probability density for populating particular filament positions, and from those by Boltzmann-inversion the free-energy profiles of the tracks (Fig. 4c). The free-energy profiles illustrate the local minima

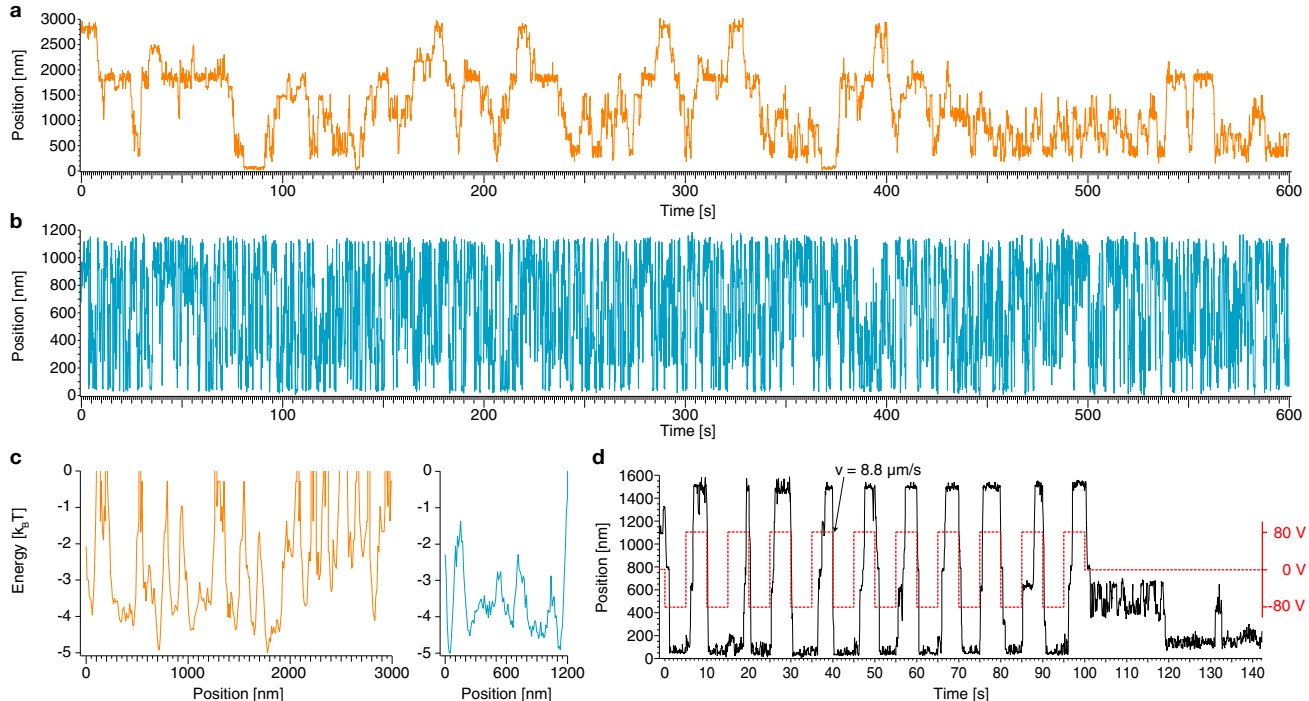

**Fig. 4 Exemplary single-particle position-time traces for motion along filaments. a, b** Exemplary single-particle traces, along the farthest measured distance (3 μm) and with the highest measured mobility, respectively. **c** Energy profiles computed from position probability distributions for the traces in **a**, **b**. **d** Red dashed line: externally applied voltage that creates an electric field along a filament. Solid line: exemplary single-particle trace of a field-driven piston.

and barriers at which particles appeared to get trapped repeatedly. For example, the highly mobile particle from Fig. 4b was confronted with few barriers, with the highest one only ~3 $k_BT$ high. By contrast, the particle from Fig. 4a which repeatedly remained stuck had more roadblocks in its way, with barriers up to 5 $k_BT$ high.

**Driving unidirectional piston motion.** To see whether external forces can drive the motion of the piston and potentially accelerate the overcoming of roadblocks, we also tracked the motion of particles in the presence of applied electric fields. Two electrodes in a previously described setup[28] were used to generate an electric field, whose direction was inverted every 5 s (Fig. 4d). We observed that particles in tracks that were oriented in parallel to the field rapidly moved in response to the field (Supplementary Fig. 37, Supplementary Movies 6 and 7). This means that when the field switched, the pistons moved quickly from one extreme of the underlying filament to the other and then remained stuck there until the field was switched into the opposite direction. When the field was switched off, the pistons showed diffusion-with-traps type behavior. These findings suggest that the trapping is caused by permanent features of the track and cannot be cleared simply by pulling the piston by force along the entire track. For pistons in tracks that were oriented perpendicularly to the field, as expected, the field had little to no effect on the motion (Supplementary Fig. 38).

**Mobility and the underlying energy landscape.** To quantify the effective (free) diffusive mobility, we computed the mean-square displacement (MSD) over time intervals from the single-particle position time traces (Fig. 5a–c). The MSD traces are first linear in time as expected for normal diffusion but then they saturate. The saturation reflects that the diffusion occurs in tracks with finite size. For our system, the

confinement length corresponds to the entire filament length, i.e., to the distance between major roadblocks. From analyzing many particles, we find that the majority of particles has diffusivities up to 0.1 μm²/s, but the fastest recorded particles that had little to no visible roadblocks moved with up to 0.3 μm²/s (Fig. 5b). The diffusive mobility increased with increasing ambient temperature, as seen by comparing the MSD from single particles recorded at 20°, 25°, 30°, and 35 °C (Fig. 5c). The diffusivity approximately doubled when going from 20 °C to 35 °C.

We computed the spatial autocorrelation of the probability density to populate track positions from each recorded particle, to investigate for hidden periodicities. The spatial autocorrelation function (Fig. 5d), averaged over many single-particle recordings, reveals clear periodic peaks occurring in intervals of 64 nm, which matches the length of a single barrel subunit. The periodicity can also be seen in the spatial autocorrelation from individual particles, albeit less clearly. The fact that the designed periodicity of the track is recovered from the motion of the particles suggests that the mobile units get momentarily trapped in periodically occurring structural features. This behavior could be caused for instance by the piston docking site (a small depression), which appears in every barrel monomer.

The hallmark of Brownian motion is a Gaussian velocity distribution. By contrast, the velocity distribution computed from the experimental random walker position-time traces deviates strongly from a Gaussian (Fig. 5e). The non-Gaussian behavior of distributions that are averaged over individual particles does not necessarily mean that the motion of a single, freely diffusing particle follows a non-Gaussian process, but could alternatively stem from deviations among individual particles[29]. In the present case, instead, we suspect that additional non-Gaussian effects come from the periodic potential landscape connected to the molecular structure of the barrel. To substantiate this further, we

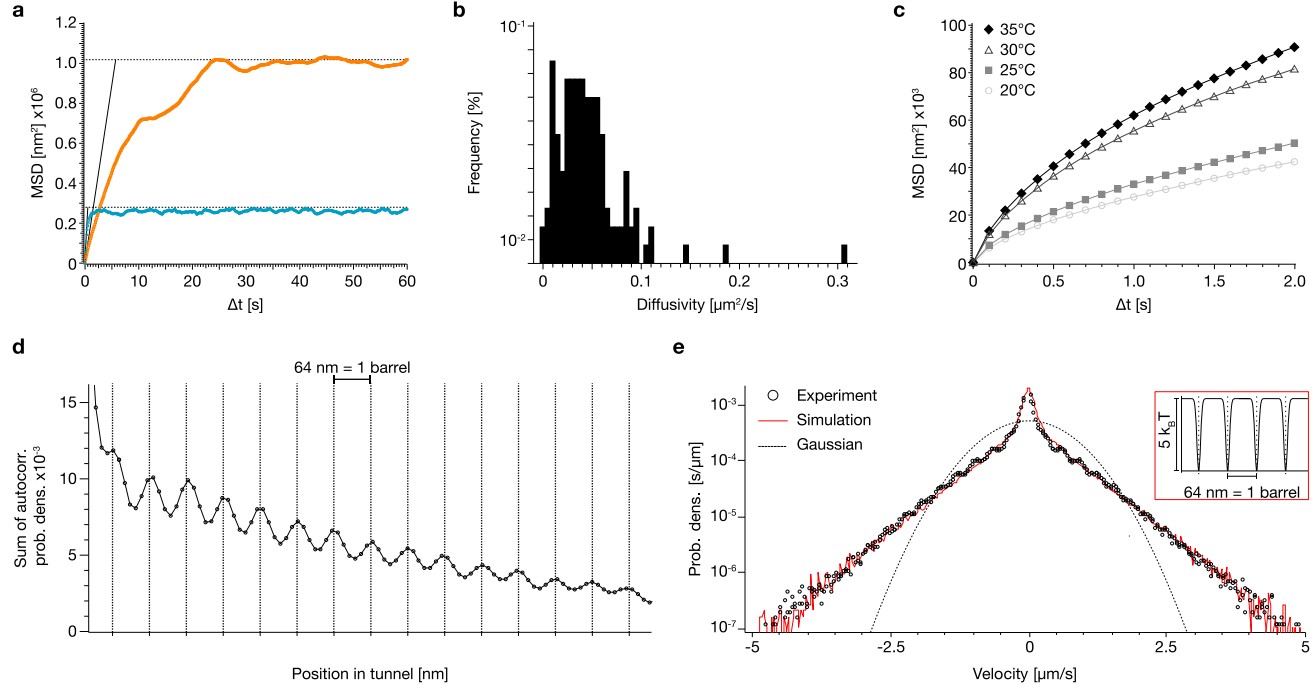

**Fig. 5 MSD, track periodicity, velocity distribution. a** Mean square displacement (MSD) curves of the single-particle traces in Fig. 4a, b, respectively (calculated according to Supplementary Software File 3). **b** Histogram of the diffusivities D of $N = 128$ particles. **c** Symbols: Mean MSD as a function of time intervals for single-particle motion, recorded at the indicated temperatures. 20 °C: $N = 28$, 25 °C: $N = 32$, 30 °C: $N = 37$, 35 °C: $N = 31$. **d** Sum of the spatial autocorrelation computed for $N = 128$ particles. **e** Circles: velocity distributions computed from the time-derivative of single-particle position-time traces. Dotted line: Gaussian distribution. Red solid line: velocity distribution simulated using Langevin dynamics (calculated according to Supplementary Software Files 1 and 2). Inset: free energy surface used in the simulation.

compare our experimental findings with Langevin simulations of a particle in a quasi-periodic potential (Fig. 5e, inset).

In the simulation model, the particle diffuses driven by random white noise that mimics the thermal stochastic environment. The particle velocity's standard deviation depends sensitively on the barrier height (see Supplementary Note 1). The analysis of the experimental traces suggests a significant variation of potential barriers (e.g., Fig. 4a vs 4b). Accordingly, we averaged simulated trajectories with potential barrier heights that vary in a range between 1 and 10 $k_B$T. To match experimental conditions, we also added localization noise to the simulated trajectory positions. With a fixed period of $a = 64$ nm and a fixed potential well width of $b = 4$ nm, the velocity distribution obtained by averaging over many thus simulated single-particle trajectories agrees favorably with the experimentally observed velocity distribution, which shows that the non-Gaussian behavior is due to the periodic potential landscapes with different barrier heights.

## Discussion
In this work, we present an artificial macromolecular transport system that is based on an inverted mechanical interlocking concept: A mobile DNA building block moves through a micrometer-long tube with mobilities up to 0.3 μm²/s, with up to 3 μm total displacements. Micrometer distances can be covered diffusively in fractions of seconds. Our artificial transport system therefore accomplishes an important mechanistic step forward: it achieves diffusivities and displacement ranges known from natural molecular motors. It realizes a mobility leap over five orders of magnitude compared to previous DNA random-walker designs (i.e., 17 nm²/s[23]). It allows tracking motions in real-time using the very same techniques that researchers used to reveal the secrets of natural protein motors. As such it presents an excellent starting

point to build and study artificial motors that move autonomously driven by chemical fuels.

The performance of our nanoscale tube system currently appears limited solely by residual stochastic defects in the track, whose occurrence can presumably be further reduced in future designs. A key ingredient in our design was protecting the mobile unit from the exterior. With this protection, our system can also enable systematic nanotribology and mobility studies, for example as a function of environmental parameters such as temperature as we discussed.

The chemical synthesis of small artificial molecular motors (AMM) including mechanically interlocked molecules such as rotaxanes and catenanes has greatly advanced the understanding of the requirements for building such objects, and how to drive directed molecular motions using chemical fuels, light, and other stimuli[30–36]. Theoretical frameworks for directing random Brownian motion of molecules or their parts were primarily developed by analyzing natural motor proteins[37–42] and by using AMMs as model systems. Our DNA-based transport system could provide a structural framework in which the smaller AMMs generated by chemical synthesis could be embedded to couple the directed motion of AMM to walker movement.

Changes in environmental parameters such as pH, ionic strength or temperature have been previously used to induce conformational changes in DNA nanostructures, based on the sensitivity of certain DNA motifs to such parameters[25,43–45]. External fields and light have also been explored for generating motion in switch-like or rotary DNA nanostructures[28,46–48]. In the present work, driven by electric fields, we realized long-range linear motion of DNA building blocks along molecular tracks with high speeds up to 9 μm/s. Controllable physical movement of matter along molecular tracks could be used to transport materials between compartments[7] and to mechanically gate the

state of larger biochemical machines such as an artificial cell, by analogy to the flow of electrons in an electric circuit. Directional in-plane electric fields could be used to drive molecular cargo transporters through molecular tracks that are connected by DNA gates with physical valves in between that may be operated by e.g., out-of-plane electric fields or by using biochemical stimuli. Gopinath et al have described how to place DNA nanostructures on solid-state surfaces in a programmable fashion[49], and Schulman et al have shown how to connect distal landmarks on surfaces with DNA nanotubes[50]. These procedures could potentially be combined with our transport system to build a molecular transportation network.

Next to the connection of distant reaction compartments, protected one-dimensional transport within a tubular track suggests a variety of other potential applications. For example, the "piston" could be used to pull other cargo into the tube. One possibility would be to pull in and stretch DNA molecules for barcoding analysis (similar as in silicon-nanofabricated channels). Aligning enzymes inside of the tube could be used to generate "assembly line"-like multi-enzyme cascades, in which the reactants (loaded on the piston) are protected from the environment by the tube and are subjected to enzymatic modifications in a strict order, dictated by the 1D geometry of the channel. To further insulate the system from chemical disturbances, it is also conceivable to coat the tubes for example with an impermeable lipid bilayer.

## Methods

**Design of scaffolded DNA origami objects**. All objects were designed using caDNAno v0.2[51].

**Folding of DNA origami nanostructures**. The folding reaction mixtures for the piston and the two capping objects contained scaffold DNA[52] at a final concentration of 50 nM and oligonucleotide strands (IDT Integrated DNA Technologies) at 200 nM each. The folding reaction mixtures for the barrel contained both scaffolds DNA[52,26], at a final concentration of 20 nM each and oligonucleotide strands (IDT Integrated DNA Technologies) at 150 nM each. The folding reaction buffers contained 5 mM TRIS, 1 mM EDTA, 5 mM NaCl (pH 8) and 15 mM MgCl₂ for the barrel and 20 mM MgCl₂ for the piston and caps. The folding reaction mixtures were subjected to different thermal annealing ramps using TETRAD (MJ Research, now Biorad) thermal cycling devices. Barrel: 15 min at 65 °C, followed by three-hour intervals for each temperature, starting at 56 °C down to 53 °C, decreasing by 1 °C every step. Piston: 15 min at 65 °C, followed by one-hour intervals for each temperature, starting at 64 °C down to 47 °C, decreasing by 1 °C every step. Caps: 15 min at 65 °C, followed by one-hour intervals for each temperature, starting at 52 °C down to 40 °C, decreasing by 1 °C every step. Finally, all folding reaction mixtures were incubated at 20 °C before further sample preparation steps. All DNA sequences are available in the Supplementary Data Files 1–4.

**Purification of DNA origami nanostructures by gel extraction**. All folded objects were purified from excess oligonucleotides by gel-electrophoretic separation[53]. The samples were electrophoresed on 1.5–2.5% agarose gels containing 0.5x tris-borate-EDTA and 5.5 mM MgCl₂ for 1–3 h at 70 or 90 V bias voltage in a water-cooled gel box. The desired bands where then cut out of the gel with an X-tracta Generation 2 hand punch. The extracted sample was then centrifuged at 2000 rcf for 5 min in a Freeze 'N Squeeze DNA Gel Extraction Spin Colum, pore size 0.45 μm (BioRad).

**Buffer exchange of DNA origami nanostructure samples by ultrafiltration**. Buffer exchange after gel extraction was performed via ultrafiltration (Amicon Ultra 0.5 ml Ultracel filters, 50 K and 100 K) with buffer containing 5 mM TRIS, 1 mM EDTA, and 500 mM NaCl[54]. All centrifugation steps were performed at 10k G for 3–10 min at 25 °C. The filters were first filled up with 0.5 ml of buffer and centrifuged. 0.5 ml of origami sample were then added and centrifuged. Another 3 rounds of adding 0.45 ml buffer and subsequent centrifugation were performed before a final retrieving step, where the filter inset was turned upside down, placed into a new tube and centrifuged.

**Negative-stain TEM**. 5 μl of sample (10–50 nM DNA object concentrations, 20–40 mM MgCl₂ concentration, and 1–4.5 M NaCl) were pipetted onto a plasma-treated (45 s, 35 mA) formvar-supported carbon-coated Cu400 grid (Electron Microscopy Sciences). The sample droplets were incubated for 30 s–10 min on the

grids and then blotted away with filter paper. A 5 μl droplet of 2% aqueous uranyl formate (UFO) solution containing 25 mM sodium hydroxide was added and blotted away as a washing step. A 20 μl UFO droplet was then added, incubated for 30–40 s and blotted away. The grids were then air dried for 10–20 min before imaging in a Philips CM100, an FEI Tecnai 120 and a Jeol JEM3200 FSC microscope. Micrographs were recorded using AMT 600 Software and SerialEM. Automated particle picking was performed with crYOLO[55], 2D class averaging was performed with Relion[56].

**JEOL JEM3200 FSC image acquisition**. The JEOL JEM3200 FSC was operated at 300 kV and the energy filter slit was set to 20 eV width. Images were acquired with an DE64 direct electron detector operated in integration mode at a magnified pixel size of 16 A. 75 frames were saved per image and subsequently corrected for residual motion with 10×10 patches using the RELION 3.0.8 implementation of Motioncor2[57].

**Gel electrophoresis**. All samples were electrophoresed on 1.5–2.5% agarose gels containing 0.5x tris-borate-EDTA and 5–25 mM MgCl₂ for 1–3 h at 70 or 90 V bias voltage in a water or ice-water-cooled gel box. The loaded samples contained final monomer concentrations of 5–20 nM (unless otherwise noted). The gels were stained with ethidium bromide, if the samples did not include fluorescent dyes and were scanned with a Typhoon FLA 9500 laser scanner (GE Healthcare) at a resolution of 25 or 50 μm/pixel. Resulting images were further processed using Photoshop CS6.

**Transport system sample preparation protocol for fluorescence measurements**.

1. Folding of all four monomers at their respective optimal folding conditions, according to initial folding screens.
2. Purification of excess oligonucleotides by gel extraction for all four samples.
3. Addition of a Biotin-modified oligonucleotide to the purified barrel sample to bind to the respective anchor sequence on the outside of the barrel.
4. Four rounds of ultrafiltration with buffer containing 0.5 M NaCl and no MgCl₂ for all four samples.
5. Mix the barrel and the piston sample at a barrel to piston ratio of 6:1 – 10:1, incubate at 30 °C in 3 M NaCl for 12–16 h to build dimers.
6. Add oligonucleotide sequences to permanently close the barrel and incubate at 30 °C for 15 min.
7. Incubate the closed barrel-piston dimers at 40 °C in 2 M NaCl for 1 h to reduce possible aggregation due to high salt conditions.
8. Add the first set of oligonucleotide sequences to trigger the polymerization of piston-loaded and empty barrels at 4:1 staple to barrel ratio. Incubate at 40 °C for 30–60 min.
9. Add the second set of oligonucleotide sequences for the polymerization reaction at 4:1 staple to barrel ratio. Incubate at 40 °C for 30–60 min.
10. Add the first cap at a cap to tunnel ratio of 6:1, assuming that the average tunnel consists of roughly ~30–40 barrels. Incubate at 40 °C in 2 M NaCl for 1 h.
11. Add the second cap at the same cap to tunnel ratio and incubate at 40 °C in 2 M NaCl for 1 h.

The entire construction protocol is complete within 20 h, starting with purified monomers (piston, barrel, and both caps) and finishing with fully functional transport systems, ready for real-time fluorescence experiments.

**Single-molecule fluorescence microscopy experiments for free diffusion**.

1. Cover slides (Sigma Aldrich) were cleaned in 2 M NaOH for 30 min and then sonicated in 2% Hellmanex, rinsed with double distilled water (ddH₂O), then sonicated in ddH₂O, again rinsed with ddH₂O and finally sonicated in ethanol (99%). All sonication steps were performed for 5 min. The slides were then dried at 70 °C for 1 h. A solution of 0.5% bioPEG-silane (solved in ethanol) with 1% acetic acid was incubated on the slides at 70 °C for 30 min. The slides were then rinsed with ddH₂O and flushed with N₂ until dry and stored protected from light[58].
2. Place a reaction chamber on top of the cover slides and wash with NeutrAvidin (0.05 mg/ml) in T50 buffer (10 mM Tris, 50 mM NaCl). Incubate for 10–15 min and wash with buffer containing 2 M NaCl.
3. Add the sample containing capped tunnels with pistons trapped inside to the reaction chamber and bind the tunnels to the pegylated glass surface via biotin-neutravidin interaction. Incubate for 10–30 min depending on the desired surface coverage.
4. Wash the reaction chamber with buffer containing an oxygen scavenging system (100 mM tris-HCl, 2 mM Trolox, 0.8% D-glucose, catalase (2000 U/ml), glucose oxidase (165 U/ml), purchased from Sigma Aldrich) and 1 M NaCl, along with invader oligonucleotide sequences at final concentrations of 1 μM to release the pistons from their initial docking sites. Incubate at room temperature for 15 min.
5. Acquire movies at 20–35 °C in a custom-built objective-type TIRFM (oil-immersion objective, 100x, apochromat, NA 1.49; Nixon with an acousto-

optical tunable filter (Pistonasus Optics)). Green (532 nm, Oxxius) and red (640 nm; Oxxius) diode lasers were used to excite the Cyanine-3 and Cyanine-5 dyes attached to the pistons and barrels. Their fluorescence signals were divided with dichroic mirrors and detected by two EMCCD cameras (Andor iXon+). Movies were acquired in 512×512 pixel format with a frame rate of 10 fps per channel. A piezo-driven sample stage (PInanoXYZ; Physik Instrumente) was used to manoeuvre to different positions on the cover slide. A custom written LabView routine was used to control all microscope components[59].

**Single-particle centroid tracking**. The piston diffusion was tracked using the ImageJ particle tracker plugin[60]. Calculating statistics of individual pistons was done using self-written batch code (Supplementary Software File 3) within IgorPro.

**Temperature control**. Temperature-controlled single-molecule TIRF measurements were performed using VAHEAT-Micro heating system (Interherence GmbH).

**Single-molecule fluorescence microscopy setup for driven diffusion**. For measurements with applied electric fields, a separate single-molecule TIRFM setup was used. The setup was custom-built on the basis of an Olympus IX71 microscope body. Three laser light sources with wavelengths 642 nm (Toptica iBeam smart, diode laser, 150 mW, Gräfelfing, Germany), 532 nm (Oxxius 532-50, diode-pumped solid-state laser, 50 mW, Lannion, France), and 488 nm (Toptica iPulse, diode laser, 20 mW) are used as excitation light source. The 488 nm excitation was not used for experiments in this project. All imaging is performed with a 100x oil immersion objective (UAPON 100xOTIRF objective, NA 1.49 oil, Olympus, Japan). The sample is supported by a piezo z-stage (Physik Instrumente (PI) GmbH, Karlsruhe, Germany). The filter cube was configured with a ZT532/640RPC dichroic mirror and a ZET532/640 (Chroma Technology, Olching, Germany) emission filter. Videos were acquired with an ORCA-Fusion Digital CMOS camera (Hamamatsu Photonics, Japan) attached to the left camera port.

The flow chamber, in which the structures were exposed to the applied fields is assembled from three parts (Supplementary Fig. 39). A top part made from $Al_2O_3$, double-sided adhesive tape 3 M 467MP (3 M Company, Maplewood, Minnesota, USA) and cover slip. The top part creates buffer reservoirs and helps to attach a custom-made plug. The plug secures 0.2 mm thick platinum wires, to which the operating voltage is applied. The adhesive tape is cut with a laser engraver (Trotec Speedy 100, Trotec Laser, Marchtrenk, Austria) to create a 50 µm high channel connecting the reservoirs when sandwiched between top part and cover slip. The applied voltage was controlled by a custom-built LabView routine that supplied control voltages to a custom-built operational amplifier to generate the final output voltage.

**Reporting summary**. Further information on research design is available in the Nature Research Reporting Summary linked to this article.

## Data availability
The data supporting the findings of this study are available either within the paper, its supplementary information files or the source data file. Real-time fluorescence movies are available from the corresponding author upon reasonable request.

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

## Acknowledgements
This work was supported by a European Research Council Consolidator Grant to H.D. (GA no. 724261), the Deutsche Forschungsgemeinschaft through grants provided within the Gottfried-Wilhelm-Leibniz Program (to H.D.), the SFB863 Project ID 111166240 TPA9 (to H.D.) and TPA8 (to F.C.S.), and SFB1449 Project (to R.R.N.). We thank Matthias Schickinger for support with the TIRF microscope and Martin Langecker for support with the setup used for electric-field driven transport.

## Author contributions
H.D. designed the research, P.S. performed research. H.K. performed simulations (Fig. 5). E.K., F.C.S. contributed instrumentation for electric-field driven transport in (Fig. 4). E.K. supported research with electric-field driven motions (Fig. 4). M.H. provided scaffold strands. M.K. performed TEM analysis. R.R.N. supervised simulations. F.C.S. supervised electric-field-driven transport experiments. All authors edited and commented on the manuscript.

## Funding

## Competing interests
The authors declare no competing interests.
