## [Peer Review File · Nature Communications]

Reviewers' Comments:

Reviewer #1:

Remarks to the Author:

Dietz et al present a very nice study that's a discrete and significant step beyond traditional DNA motor based systems, achieving orders of magnitude speed-up in motion using a sophisticated tubular DNA origami design with an internal piston. Rigorous characterization is performed of both the self-assembly/structure and dynamics of the system, using TEM and time-resolved single-molecule fluorescence imaging. Simulations add a useful, quantitative angle to describe how periodic traps within the cylinders result in the observed diffusive behavior. I'd normally have suggestions for improvement of this work, but it's very well designed and executed, without any obvious need for significant revision or improvement. One very minor point is in Figure 5C, MSD is typically presented as a function of time-lag, not time. And some of the fonts in this Figure may also benefit from enlargement. Finally, if there's any insight that could be provided in the main text on (1) why some of the earlier designs failed to robustly/rapidly release the piston, that would be interesting to read about; and (2) if the typical time-scale needed to initiate piston motion in response to the invader strand could be mentioned, that would also be of interest to the reader.

Reviewer #2:

Remarks to the Author:

The paper describes a physically simple but technically very impressive system of pistons in tubes, all constructed from DNA origami. Pistons move diffusively and can be driven by the application of an external electric field. The appeal of the paper lies in the elegance of the construction and the characterization of motion. The significant claim made by the authors is that, in comparison with other DNA nanosystems for constrained diffusive motion demonstrated so far, motion is very fast. To achieve directed motion in such a system would be a huge achievement - this is a good first step.